# Efficient Production of Self-Assembled Bioconjugate Nanovaccines against *Klebsiella pneumoniae* O2 Serotype in Engineered *Escherichia coli*

**DOI:** 10.3390/nano14080728

**Published:** 2024-04-21

**Authors:** Yan Zhang, Peng Sun, Ting Li, Juntao Li, Jingqin Ye, Xiang Li, Jun Wu, Ying Lu, Li Zhu, Hengliang Wang, Chao Pan

**Affiliations:** 1College of Food Science and Technology, Shanghai Ocean University, No. 999 Hucheng Huan Road, Lingang New City, Shanghai 201306, China; 2State Key Laboratory of Pathogen and Biosecurity, Beijing Institute of Biotechnology, No. 20 Dongdajie Street, Fengtai District, Beijing 100071, China

**Keywords:** self-assembling, bioconjugate nanovaccines, *Klebsiella pneumoniae* O2 serotype, mutation

## Abstract

Nanoparticles (NPs) have been surfacing as a pivotal platform for vaccine development. In our previous work, we developed a cholera toxin B subunit (CTB)-based self-assembled nanoparticle (CNP) and produced highly promising bioconjugate nanovaccines by loading bacterial polysaccharide (OPS) in vivo. In particular, the *Klebsiella pneumoniae* O2 serotype vaccine showcased a potent immune response and protection against infection. However, extremely low yields limited its further application. In this study, we prepared an efficient *Klebsiella pneumoniae* bioconjugate nanovaccine in *Escherichia coli* with a very high yield. By modifying the 33rd glycine (G) in the CNP to aspartate (D), we were able to observe a dramatically increased expression of glycoprotein. Subsequently, through a series of mutations, we determined that G33D was essential to increasing production. In addition, this increase only occurred in engineered *E. coli* but not in the natural host *K. pneumoniae* strain 355 (Kp355) expressing OPS_KpO2_. Next, T-cell epitopes were fused at the end of the CNP(G33D), and animal experiments showed that fusion of the M51 peptide induced high antibody titers, consistent with the levels of the original nanovaccine, CNP-OPS_KpO2_. Hence, we provide an effective approach for the high-yield production of *K. pneumoniae* bioconjugate nanovaccines and guidance for uncovering glycosylation mechanisms and refining glycosylation systems.

## 1. Introduction

Klebsiella pneumoniae can cause various infectious diseases, such as pneumonia, sepsis, bacteremia, meningitis, purulent liver abscesses, and urinary tract infections. It has attracted extensive attention due to its widespread presence and high drug resistance [1]. According to reports, the proportion of drug resistance of the species is as high as 35.2%, and this percentage is constantly increasing [2]. In particular, the emergence of carbapenem-resistant *Klebsiella pneumoniae* (CRKP) poses a tremendous threat to human health [3,4,5].

Vaccinations have emerged as the most cost-effective and potent tools in the prevention and control of infectious diseases. Generally, bacterial vaccines are segregated into whole-cell vaccines, protein vaccines, and polysaccharide vaccines, primarily based on their antigen types. Among them, polysaccharides such as O-polysaccharides (OPSs) and capsular polysaccharides (CPSs) that are present on bacterial surfaces are considered ideal antigen targets due to their robust specificity [6,7]. However, polysaccharides alone are T-cell-independent antigens (TI-antigens), which evoke an inefficient immune response in the absence of T-cell participation [8,9]. Notably, it was discovered that conjugate vaccines synthesized through coupling polysaccharides with carrier proteins converted these TI-antigens into T-cell-dependent antigens (TD-antigens), which could trigger efficient immune responses and foster immune memory [10]. As such, conjugate vaccines are regarded as the most successful bacterial vaccines, and multiple products (e.g., Prevnar, PCV3, and Hiberix) have been launched that now occupy a substantial market share.

The conjugated vaccines currently on the market are synthesized by chemical methods involving the extraction of polysaccharides and carrier proteins, followed by activation, chemical crosslinking, and, ultimately, purification to produce the final vaccine product [8]. With the development of synthetic biology and the discovery of glycosylation in bacteria [11], a novel biosynthetic technology has emerged: protein glycan coupling technology (PGCT) [12,13]. PGCT mainly refers to the direct coupling of polysaccharides to carrier proteins within host bacteria through a reaction catalyzed by glycosyltransferase. This system requires four essential components: engineered host cells, carrier proteins with a glycosylation motif, glycosyltransferases, and polysaccharides [14]. This method solves many shortcomings associated with chemical methods, including low yields and complex multi-step processes. The technology also permits the utilization of attenuated pathogenic bacteria as hosts to generate their corresponding vaccines, avoiding the difficulty of cloning large polysaccharide synthesis gene clusters [15]. Currently, three glycosyltransferases (PglB, PglL, and PglS) have been identified as viable options for conjugate vaccine biosynthesis [16]. PglB, the first discovered glycosyltransferase, recognizes numerous pathogenic bacterial polysaccharides [17,18,19]. The development of the PglL system further broadened the recognition range for polysaccharides [13,19,20,21]. Notably, the recently discovered PglS recognizes the *S. pneumoniae* CPSs, thereby expanding the application of this biotechnology [22]. Despite the fact that PGCT was only developed over the past decade or so, it has undergone rapid development, and multiple products have progressed to clinical trial phases [13].

Compared to chemical methods, a bioengineering-based synthesis offers a multitude of avenues for optimizing conjugate vaccines. To improve the vaccine yield, diverse strategies have been explored, such as establishing an engineered cell chassis suitable for exogenous polysaccharide synthesis by knocking out the host’s own polysaccharide synthesis genes [23]. Moreover, modifications to glycosyltransferase PglB were found to increase the glycoprotein yield and broaden its polysaccharide recognition range [24]. Furthermore, carriers significantly influence the immune response, and some toxins, such as CRM197, tetanus toxoid (TT), recombinant pseudomonas aeruginosa exotoxin A (rEPA), and diphtheria toxoid (DT), are often used to invoke potent immune responses [25].

The continuous application of nanotechnology is increasingly highlighting the advantages of carrier designs [26,27]. Studies have discovered that particles ranging between 15 and 100 nm most suitably target lymph nodes, eliciting efficient immune responses [26,28,29,30,31]. Proteinaceous nanocarriers, such as ferritin and virus-like particles (VLPs), have been extensively investigated due to their high safety and biocompatibility for prophylactic vaccines [32,33,34]. Utilizing these carriers, several high-efficiency candidate vaccines against pathogens have already been developed. Notably, a modular self-assembled nanocarrier was developed in recent years that demonstrated exemplary immune-enhancement attributes [35,36]. By fusing a pentameric cholera toxin B subunit (CTB) with a trimer-forming peptide, the monomers (CTBTri) self-assembled into nanoparticles (CNPs) during expression [37]. Moreover, through integration with PGCT, a range of pathogenic bioconjugate nanovaccines have been prepared, showcasing superior humoral and cellular-immune responses. Additionally, a dry powder inhalation vaccine for COVID-19 based on CNPs showed the potential to stimulate an intense mucosal immune response [38]. Such results underscore the formidable application potential of this self-assembled nanocarrier.

For *K. pneumoniae*, both the OPS and CPS can serve as antigens. However, the OPS may be more suitable for vaccine design as it has only eight serotypes, with approximately 80% of the clinical isolates belonging to one of the four serotypes (O1, O2, O3, and O5), while the CPS has over 77 serotypes, of which 25 account for almost 70% of the clinical isolates. In particular, researchers have also found that serotype O2 has a selective advantage against antibiotics and is becoming the predominant serotype in the isolated extended-spectrum beta-lactamase and carbapenem-resistant Enterobacteriaceae subgroups, with proportions as high as 35% and 50%, respectively [1]. Therefore, *K. pneumoniae* vaccines targeting the O2 polysaccharide can provide better coverage against MDR isolates than vaccines targeting other *K. pneumoniae* antigens.

In our previous studies, we constructed engineered *E. coli* W3110Δ*waaL*Δ*wbbH-L* and developed a *K. pneumoniae* O2 serotype bioconjugate nanovaccine using a CNP as the carrier, which demonstrated superior efficacy compared to traditional vaccines [23]. However, low production limited the further application of this vaccine. In this study, we discovered that mutating the 33rd amino acid glycine (G) in the CNP into aspartate (D) significantly boosted the production of glycoproteins. It was further unveiled that the G33D mutation was the sole condition for this effective glycosylation at and around position 33 across a range of mutations. Yet, this notable efficiency improvement was not observed in the endogenous host Kp355 expressing *K. pneumoniae* O2 polysaccharide (OPS_KpO2_). Subsequently, by fusing T-cell epitopes at the end of the CNP(G33D), the new vaccines induced a high antibody titer, consistent with levels of the original bioconjugate nanovaccine CNP-OPS_KpO2_, and also had a higher yield. Consequently, we have formulated a recombinant strain focusing on the high-yield preparation of *K. pneumoniae* bioconjugate nanovaccines. Although the mechanism of increased production is inadequately fully explained by a single factor, we propose a potential hypothesis, diverging from the mainstream understanding, on high-yield glycoprotein acquisition and provide guidance for uncovering glycosylation mechanisms and refining glycosylation systems.

## 2. Materials and Methods

### 2.1. Strains and Plasmids

*E. coli* W3110Δ*waaL*Δ*wbbH-L* (W3110ΔΔ) is an *E. coli* W3110 strain in which the *waaL*, *wbbH*, *wbbI*, *wbbJ*, *wbbK*, and *wbbL* genes are deleted. Kp355Δ*waaL* is the O2 serotype of *K*. *pneumoniae* strain 355 with the *waaL* gene deleted. 50973DWC/CldLT2 is *Salmonella paratyphoid* A CMCC strain 50973 with the *waaL* and *cld* genes deleted and the *S. typhimurium cld* gene (*cld*LT2) inserted. All strains were stored in our laboratory. *E. coli* DH5α was purchased from TransGen Biotech.

Plasmids pACYC184-OPS_KpO2_, expressing the O2 serotype OPS of *K. pneumoniae*, pACYC184-OPS_KpO1_, expressing the O1 serotype OPS of *K. pneumoniae*, pET28a-pglL-NP, expressing the glycosyltransferase PglL and self-assembled nanocarrier CNP, and pACYC184tac-OPS_Ba_, expressing the O:9 serotype OPS of *Yersinia enterocolitica* strain 52212, were stored in our laboratory. Other base-mutated plasmids such as pET28a-pglL-CNP(G33X) (X is an amino acid (alanine (A), aspartic acid (D), glutamic acid (E), phenylalanine (F), histidine (H), isoleucine (I), lysine (K), leucine (L), methionine (M), asparagine (N), proline (P), glutamine (Q), arginine (R), serine (S), threonine (T), valine (V), tryptophan (W), or tyrosine (Y)) and pET28a-pglL-CNP(A32D, K34D, in33D, or in34D) were constructed in this study. The plasmids used in this study are listed in Table 1.

### 2.2. Bacterial Culture

All strains were cultured in an LB medium (0.5% yeast extract, 1% sodium chloride, and 1% tryptone) or solid medium (LB with added 1.5% agar) for 8–12 h at 37 °C. For protein expression, strains carrying expression plasmids were cultured at 37 °C until the OD_600_ reached 0.5–0.6. Then, isopropyl-β-D-thiogalactopyranoside (IPTG) was added to a final concentration of 1 mmol/L, and cells continued to culture at 30 °C for about 12 h.

### 2.3. Base Mutation

The base in the plasmid was mutated using a Fast Site-Directed Mutagenesis kit (Tiangen, Beijing, China), and the primers are shown in Appendix A. After thoroughly mixing the primers, plasmid (pET28a-pglL-CNP), DNA polymerase, and ddH_2_O, the PCR amplification was performed. Subsequently, the PCR product was mixed thoroughly with the DpnI restriction enzyme, and the reaction was carried out at 37 °C for 1 h. Afterward, 5 µL of the DpnI digestion product was transferred into *E. coli* DH5α cells and cultured onto the surface of the LB solid medium containing corresponding antibiotics for selection. After incubating at 37 °C for 12 h, single clones were picked for sequencing.

### 2.4. Immunoblotting

Immunoblotting was performed as described previously [23]. Briefly, 1 mL of the overnight-induced bacteria was centrifuged. The bacterial pellet was resuspended in ddH_2_O. Then, the suspension was mixed with an equal volume of 2 × SDS buffer (100 mM Tris-HCl, pH 6.8, 3.2% *w v*^−1^ SDS, 0.04% *w v*^−1^ bromophenol blue, 16% *v v*^−1^ glycerol, and 40 mM DL-dithiothreitol), thoroughly vortexed, and then boiled for 10 min in a water bath. Proteins from the supernatant were separated using SDS-PAGE and then transferred onto a pre-activated polyvinylidene fluoride (PVDF) membrane using a rapid wet transfer apparatus. Following this, the membrane was blocked by incubating in a blocking solution, tris-buffered saline, and Polysorbate 20 (TBST) that contained 5% skimmed milk powder at 37 °C for 1 h. Afterward, it was washed three times with TBST, with each wash lasting 7 min. The membrane was then incubated with horseradish peroxidase (HRP) labeled 6 × His antibody (1:1500, Abmart, Shanghai, China) at room temperature for 1 h. For polysaccharide detection, a KpO2-specific serum (1:15,000) was used as the primary antibody, and the membrane was subsequently incubated with an HRP-labeled goat anti-rabbit antibody (1:5000, Transgen Biotech, Beijing, China). After washing three times again, a prepared super-sensitive ECL luminescence reagent (Biodragon, Suzhou, China) was added to the membrane, and the results were imaged using a detection system (ChemiDoc MP Imaging System, Bio-Rad, Hercules, CA, USA).

### 2.5. Glycoprotein Purification

Cells were induced overnight, collected by centrifugation, and resuspended with buffer A1 (0.5 M NaCl, 10 mM imidazole, and 20 mM Tris-HCl, pH 7.5). Then, the cells were disrupted using a high-pressure homogenizer (Ph.D.), and the supernatant was collected through centrifugation. The supernatant was subsequently loaded onto a Ni affinity column (Roche, Basel, Switzerland), which was pre-equilibrated with A1. Then, buffer B (0.5 M NaCl, 0.5 M imidazole, and 20 mM Tris-HCl, pH 7.5) was used to elute the glycoproteins. After confirming the glycoproteins through Coomassie blue staining, the eluted solution was concentrated using ultrafiltration. Then, the product was loaded onto a Superdex 200 prep grade column (GE Healthcare, Chicago, IL, USA) using a phosphate-buffered saline (PBS) as the mobile phase at 1 mL/min. The protein solution was collected sequentially in a 2 mL tube and verified through Coomassie blue staining.

### 2.6. Experimental Animals

Six- to eight-week-old pathogen-free BALB/c mice (female) were purchased from SPF Biotechnology (Beijing, China) and housed at the Laboratory Animal Centre of the Academy of Military Medical Sciences. All animal experiments were approved by the Academy of Military Medical Sciences Animal Care and Use Committee (Approval Code: IACUC-DWZX-2020-042).

### 2.7. Immunization Experiments

Mice were subcutaneously immunized with PBS, OPS_KpO2_, CNP-OPS_KpO2_, CNP(G33D)-OPS_KpO2_, or M51-CNP(G33D)-OPS_KpO2_ on days 0, 14, and 28. Apart from the PBS group, each injection contained 2.5 µg of polysaccharide. Ten days after the last immunization, blood was sampled from the tail tip, and the serum was subsequently isolated for future analysis. Antibody titers in the serum were measured using an enzyme-linked immunosorbent assay (ELISA). Fourteen days following the third immunization, the mice were intraperitoneally injected with Kp355, and the amount of bacteria in the lungs was determined 2 days later.

### 2.8. ELISA

Ninety-six well-plates were coated with lipopolysaccharides (LPS) (10 µg/well), which were diluted in 50 mmol of Na_2_CO_3_-NaHCO_3_ (pH 9.6). After incubating at 4 °C overnight, the plates were washed three times with PBST (PBS with 0.05% Tween 20). Then, 200 µL of the blocking solution (PBST with 5% skimmed milk) was added to each well and incubated at 37 °C for 2 h. After washing, a diluted serum was added to each well, and the plates were incubated at 37 °C for one hour. After washing and drying again, 100 µL of HRP-conjugated goat anti-mouse IgG antibody (Abcam, AB6820, 1:10,000) was added to each well. After incubating and washing again, the plates were treated with 100 µL of 3,3′,5,5′-Tetramethylbenzidine (TMB) solution for a color reaction. Based on the degree of color rendering, 50 mL/well of termination solution (2M H_2_SO_4_) was added to stop the reaction. Then, a microplate spectrophotometer was used to read the plate at an optical wavelength of 450 nm [23].

### 2.9. Statistical Analysis

Data are presented as the mean ± s.d. Statistical analysis was performed by using GraphPad Prism 8.0 software. For multiple-group comparisons, a one-way ANOVA with Dunnett’s multiple comparison test was used. Statistically significant differences are indicated as * *p* < 0.05, ** *p* < 0.01, *** *p* < 0.001, **** *p* < 0.0001; ns indicates not significant (*p* > 0.05).

## 3. Results

### 3.1. G33D Mutation of CTBTri Dramatically Increased the Expression of Glycoprotein

In our previous work, we constructed a Nano-B5 platform for the biosynthesis of self-assembled proteinaceous NPs for vaccination, and highly efficient vaccine products were produced. In particular, we produced a *K. pneumoniae* bioconjugate nanovaccine (Figure 1A), which exhibited high safety and efficacy against various infections. However, low production limited the application of this vaccine. To address the problem, we hoped to obtain high-yield strains by modifying the carrier protein. We first transferred the constructed plasmid pET28a-pglL-CNP(G33D) into the host cell W3110Δ*waaL*Δ*wbbH-L/*pACYC184-OPS_KpO2_ (W3110ΔΔ/KpO2) while using the original plasmid pET28a-pglL-CNP as a control. The recombinant strains W3110Δ*waaL*Δ*wbbH-L*/pACYC184-OPS_KpO2_, pET28a-pglL-CNP and W3110Δ*waaL*Δ*wbbH-L*/pACYC184-OPS_KpO2_, and pET28a-pglL-CNP(G33D) were cultured in an LB medium and induced with IPTG overnight. Then, the whole protein from each strain was separated by SDS-PAGE and analyzed by Coomassie blue staining and immunoblotting using an HRP-labeled 6 × His antibody. To our surprise, the quantity of glycoprotein CNP(G33D)-OPS_KpO2_ in W3110Δ*waaL*Δ*wbbH-L*/pACYC184-OPS_KpO2_,pET28a-pglL-CNP(G33D) increased dramatically compared to that in the original strain, as shown in the western blotting result (Figure 1B). Subsequently, we purified CNP(G33D)-OPS_KpO2_ using affinity and size-exclusion chromatography, which was consistent with the CNP-OPS_KpO2_ purification strategy. Coomassie blue staining and immunoblotting results showed consistent, typical glycosylation ladder-like bands between CNP(G33D)-OPS_KpO2_ and CNP-OPS_KpO2_ (Figure 1C). Next, we analyzed the two glycoproteins through dynamic light scattering, and the results showed that both glycoproteins were uniformly monodispersed, with a size of approximately 20 nm (Figure 1D,E).

### 3.2. Analysis of Glycosylation of the Various Mutations at the 33rd Position in the CNP

Encouraged by the result above, we further mutated the 33rd G to 17 other amino acids, including A, E, F, H, I, K, L, M, N, P, Q, R, S, T, V, W, and Y, to reveal the relationship between this site and glycosylation (Figure 2A). After mutating the CNP, the corresponding 17 plasmids were sequenced to determine the correct mutations (Figure 2A). They were then transferred into the host W3110ΔΔ/KpO2 and induced with IPTG overnight. Western blotting results showed that all mutated carrier proteins were detected. However, although some mutations (such as G33Q and G33S) increased the expression of glycoprotein compared to the original sequence, the increases were far less than that of G33D (Figure 2B). We initially speculated that the negative charge of D may contribute to the glycosylation, but the results revealed that all other negatively charged amino acids (such as E) seemed to have no obvious effects. Even E, which was only slightly longer than D and S, with a similar structure (Figure 2C), did not show the significant changes in glycosylation as those of the G33D mutant. Therefore, we hypothesized that the 33rd D, as the only suitable amino acid to achieve a high production of glycoproteins, may be influenced by multiple factors.

### 3.3. Evaluation of the Glycosylation by Mutating Animo Acids to D around the 33rd Position

Previous results showed that D seemed the only suitable amino acid at the 33rd position for glycosylation in the system. Due to the distance between this region and the glycosylation site, it was possible that the local spatial structure caused by D was more suitable for exposing the glycosylation site. Thus, we further mutated the amino acids at the N and C termini of the 33rd amino acid G. We first mutated the 32nd amino acid A or 34th amino acid K to D from the original plasmid CNP (Figure 3A). After induction and cultivation as described above, whole-cell lysate samples were analyzed through Coomassie blue staining and immunoblotting. Western blotting results showed that both A32D and K34D had no effect on enhancing glycosylation, although a slight improvement was observed with each mutation compared to the original strain (Figure 3B). In order to minimize the impact on the local structure around the 33rd amino acids, we then inserted a D before and after the 33rd amino acids (Figure 3C). After confirming the correctness of the mutation through sequencing, subsequent immunoblotting results indicated that these mutations also did not promote glycosylation efficiency (Figure 3D). Therefore, we speculated that only the G33D mutation in this local sequence facilitated glycosylation.

### 3.4. Glycosylation Analysis of CNP(G33D) for Various OPSs

After determining the local uniqueness of G33D in the carrier, we further analyzed the impact of polysaccharide types on glycosylation efficiency. In our previous research, we established glycosylation systems in a series of pathogens and prepared candidate conjugate vaccines. The plasmids pET28a-pglL-CNP and pET28a-pglL-CNP(G33D) were introduced into 50973DWC/CldLT2, Kp355Δ*waaL*, W3110Δ*waaL*Δ*wbbH-L/*pACYC184-OPS_KpO1_ (named W3110ΔΔ/O1), and W3110Δ*waaL*Δ*wbbH-L/*pACYC184tac-OPS_Ba_ (named W3110ΔΔ/O9) (Figure 4A). After induction, the whole-cell lysis samples were analyzed, and the western blotting results showed that the coupling efficiency of CNP(G33D) with 50973DWC/cldLT2 polysaccharides sharply decreased (almost undetectable) (Figure 4B). In contrast, there was a slight increase in the binding with O9 polysaccharides; however, the carrier did not show a significant improvement in the glycosylation efficiency (Figure 4B). In addition, the expression of CNP-OPS_KpO1_ was too low to detect whole-cell samples through western blotting, and the G33D mutation was unable to improve the expression (Figure 4B). Therefore, the polysaccharide types also had a significant impact on glycosylation efficiency.

### 3.5. Analysis of Glycosylation of CNP(G33D) in Kp355

Generally, the expression efficiency of pathogenic polysaccharides in their own host is higher than in heterologous *E. coli.* However, when we introduced pET28a-pglL-CNP and pET28a-pglL-CNP(G33D) into Kp355Δ*waaL*, which expresses KpO2, we were surprised to find that no glycoprotein was detected (Figure 5). Due to the fact that the O2 gene cluster in pACYC184-OPS_KpO2_ was cloned from the Kp355 strain and worked well in W3110ΔΔ/KpO2, we speculated that the efficient glycosylation caused by the G33D mutation was also related to the host bacteria.

### 3.6. Analysis of Glycosylation of CNP(G33D) Fused with Different T-Cell Epitopes

The results above suggested that we made significant breakthroughs in the production of the *K. pneumoniae* vaccine that were conducive to further application of the vaccine. However, according to reports, mutations in G33D in CTB can prevent it from binding to the cell surface GM1, potentially reducing the intensity of the immune response. Our results also showed that the serum titer of the nanovaccine containing CTB decreased with the G33D mutation. To address this potential impact, we further fused different T-cell epitopes including P30 (TT^947−967^: FNNFTVSF WLRVPKVSASHLE), PADRE (AKFVAAWTLKAAA), M51 (M5^17−31^: LDKYELENHDLKTKN), and HA1 (HA^307−319^: (PKYVKQNTLKLAT) at the N terminus of the carrier protein. The constructed plasmids were introduced into W3110ΔΔ/KpO2, and the glycoprotein expression was analyzed. Western blotting results showed that although the amount of glycoprotein decreased when CNP(G33D) was fused with the T-cell epitope compared to that of unfused CNP(G33D), the decrease in CNP(G33D) fused with M51 (M51-CNP(G33D)) was the lowest (Figure 6A). In addition, the yield of all glycoproteins was higher than using CNP as a carrier. We fused M51 at the C terminus of CNP(G33D), adjacent to the glycosylation modification motif, and found that the expression of glycoproteins was almost undetectable (Figure 6B), suggesting that the proximity may have an impact on the spatial structure of the glycosylation motifs.

### 3.7. Evaluation of the Efficacy of Various Nanovaccines

After obtaining M51-CNP(G33D)-OPS through affinity and size-exclusion chromatography, we analyzed the antibody responses to the different nanovaccines. Briefly, BALB/c mice were immunized subcutaneously with PBS, CNP-OPS, CNP(G33D)-OPS, or M51-CNP(G33D)-OPS on days 0, 14, and 28, with each injection containing 2.5 µg of polysaccharide. Seven days after the last immunization, the blood from each mouse was sampled, and the serum was collected (Figure 7A). As expected, the ELISA results showcased a decrease in the IgG titers against the Kp355 LPS when CNP was mutated to CNP(G33D). However, we also observed that by fusing the M51 peptide at CNP(G33D), the antibody titer of M51-CNP(G33D)-OPS was consistent with that of CNP-OPS but with a higher yield (Figure 7B). Fourteen days after the last immunization, the mice were injected intraperitoneally with 2.5×10^7^ CFU of Kp355. Two days post-injection, the mice were dissected, and their lung bacterial load was determined. The results showed that although the G33D-mutated vaccine caused mice to nearly lose their ability to clear organ bacteria, the fusion of the M51 epitope restored this ability to a level consistent with the original bioconjugate nanovaccine CNP-OPS_KpO2_ (Figure 7C).

## 4. Discussion

In our previous studies, we prepared a CNP coupled with multiple antigens that exhibited efficient immune responses. In particular, coupling the CNP with the *K. pneumoniae* O2 polysaccharide induced a significant protective capability against *K. pneumoniae* infections. To scale up vaccine production, in this study, we introduced the G33D mutant of CNP into the system, which yielded an increased expression of glycoprotein and increased glycosylation efficiency. However, using a suite of mutations, we found that no other amino acid variants at the 33rd position enhanced glycoprotein expression. Also, mutations in the vicinity of the 33rd amino acid position failed to improve the glycoprotein yield, signifying G33D as the sole mutation that increased glycosylation in that particular position. Furthermore, upon analyzing the glycosylation of CNP(G33D) across different host organisms, we discerned that none substantially improved the glycoprotein yield. Surprisingly, within the host strain Kp355 (expressing the *K. pneumoniae* O2 polysaccharides), G33D led to a substantial decline in glycosylation to almost undetectable levels—a stark contrast to the outcomes observed in the host W3110. Consequently, our research provides an effective approach for the high-yield production of bioconjugate nanovaccines against the *K. pneumoniae* O2 serotype (Figure 8) and reveals a series of factors influencing protein glycosylation, thereby offering directions for glycosylation mechanism analysis and the design of efficient carriers.

In the biosynthesis of conjugate vaccines, glycosylation efficiency significantly determines the sugar-to-protein ratio and polysaccharide antigen loading rate. Despite using an identical glycosylation modification motif, different carrier proteins may present variable glycosylation efficiencies [39]. This variability is also evident in reported O-linked glycosylation systems constructed in *Shigella*, showcasing differing efficiencies across three distinct carrier proteins (CTB and rEPA) [15]. Generally, the glycosylation modification motif has been fused at the N or C terminus of carrier proteins to form specific spatial conformation during expression, thus facilitating recognition by glycosyltransferases. However, due to the proximity, the carrier protein may impact the spatial conformation of the fused motif. Moreover, mutations in the carrier protein can also result in conformational alterations in the glycosylation motif. Our research discovered that the G33D mutation in the self-assembled CNP markedly boosted glycosylation efficiency. In contrast, other mutations at or around position 33 failed to show the same effectiveness. This suggested that the G33D-induced conformational changes might precisely render the glycosylation modification motif fully recognizable by the glycosyltransferase PglL. Although the distance between the 33rd amino acid and the glycosylation modification motif in the nanostructures remains unknown, a direct or indirect association appears inevitable. These findings on mutation strategies offer insights into optimizing glycosylation efficiency.

The most notable impact of the G33D mutation is the alteration of local charges. This shift implies that the negative charge of D might attract molecules bearing positive charges. Nonetheless, the presence of numerous negatively charged amino acids on the nanoparticle surface and the mutation of the 33rd amino acid to E—another negatively charged amino acid—yielded no discernible effect. Further analysis of the CTB structure revealed that this 33rd amino acid played a crucial role in forming a pocket typically associated with GM1 binding [40]. We theorized that following the mutation from G to D, although the pocket structure remained intact [41], the charge differential was altered. This alteration potentially attracted a positively charged molecule that fits the pocket structure, resulting in spatial conformational changes to the nanoparticle. This could subsequently trigger conformational changes in the glycosylation modification motif, thereby promoting glycoprotein expression. The reduction in glycosylation efficiency observed in Kp355 expressing the *K. pneumoniae* O2 polysaccharide may be because of the absence of such fitting molecules. These hypotheses on the glycosylation efficiency changes require further experimental validation, notwithstanding their explanatory potential.

Polysaccharides significantly impact glycosylation, as revealed by our findings that both CNP and CNP(G33D) exhibited variable coupling efficiencies with various polysaccharides. This phenomenon was also observed in N-linked glycosylation research [17]. Generally, the structures of polysaccharides are complex, and their synthesis requires multiple enzymes performing catalysis in a certain order [6,42]. Many rate-limiting steps and complex external environments can vary the yield of each polysaccharide. Notably, the glycosylation efficiency was higher in the native host than in *E. coli* (expressing exogenous polysaccharides), suggesting that these enzymatic reactions are more suitable in their native host environment. Interventions in the sugar synthesis pathway can also increase yields [39]. Additionally, while polysaccharide antigens are almost always uncharged, their lengthy, intricate structures and spatial conformations potentially influence polysaccharide recognition by glycosyltransferases.

At present, there are two main strategies for the biosynthesis of conjugate vaccines: in their natural hosts or in engineered *E. coli*. Despite the higher efficiency in the native host, a universal *E. coli* cell might serve as a helpful chassis for synthesizing vaccines aimed at various pathogens [13]. This approach could prove beneficial for industrialization and quality control. Nevertheless, the major hindrance to heterologous polysaccharide expression in *E. coli* is the cloning of extensive polysaccharide gene clusters and ensuring efficient polysaccharide expression. The insufficient synthesis efficacy of heteropolysaccharides restricts the yield of the final product. Our study introduced a novel mutation method to enhance glycoprotein expression, in contrast to conventional techniques such as host modification, glycosyltransferase optimization, and glycosylation motif modification.

CTB is usually recognized as a mucosal adjuvant due to its ability to bind to GM1 on the surface of mucosal epithelial cells, thereby facilitating antigen endocytosis [43,44,45]. However, the G33D mutation deprives this binding capacity [41]. Given the widespread presence of GM1 on antigen-presenting cells, binding with GM1 may also play a role in boosting the immune response. Although the G33D mutation may potentially reduce the vaccine’s immunogenicity, a substantial enhancement in the production yield is advantageous for ongoing optimization and production. Further enhanced immune responses could be obtained by adding T-cell epitopes, appropriate adjuvants, or delivery vectors [27,38,46,47]. Among them, many vaccines have achieved good immune effects by adding appropriate epitopes. For example, it has been confirmed that the tumor-associated glycoprotein mucin1 (MUC1) was coupled with T-cell epitopes, and this vaccine induced high-titer specific antibodies in animal experiments [48,49,50]. In addition, by coupling tumor-related MUC1 glycopeptides and T-helper-cell epitopes with polymer-carriers, the immunogenicity of anti-tumor vaccines can be enhanced [50,51]. Similarly, many T-cell epitopes have also been used in the research of tuberculosis vaccines and have shown the ability to enhance immune efficacy in mice [52]. Because we want to develop an adjuvant-free vaccine, we first considered fusing antigenic epitopes. We screened a high-yield vector M51-CNP(G33D), and preliminary immune results found that the immune effect of this constructed nanovaccine reached the level of CNP-OPS. Subsequently, the vaccine’s efficacy will be further evaluated.

In conclusion, we have identified specific nanocarrier mutations that enhanced the efficacy of *K. pneumoniae* bioconjugate nanovaccine synthesis in W3110ΔΔ/KpO2. Afterward, through a series of further mutations, we found that only G33D significantly increased glycoprotein production. Interestingly, we found that this increase in production in *E. coli* was not detected in K. pneumoniae expressing O2 polysaccharides. Mice experiments indicated that although this mutation leads to a decrease in the immune response, we found that by adding T-cell epitopes, the immune effect can be improved to be consistent with previous CNP-OPS. Therefore, in addition to providing a high-yield approach for creating bioconjugate nanovaccines, our study discloses a succession of significant factors bearing on protein glycosylation. Such insight offers valuable implications for the analysis of glycosylation mechanisms and the design of efficient carriers.

## Figures and Tables

**Figure 1 nanomaterials-14-00728-f001:**
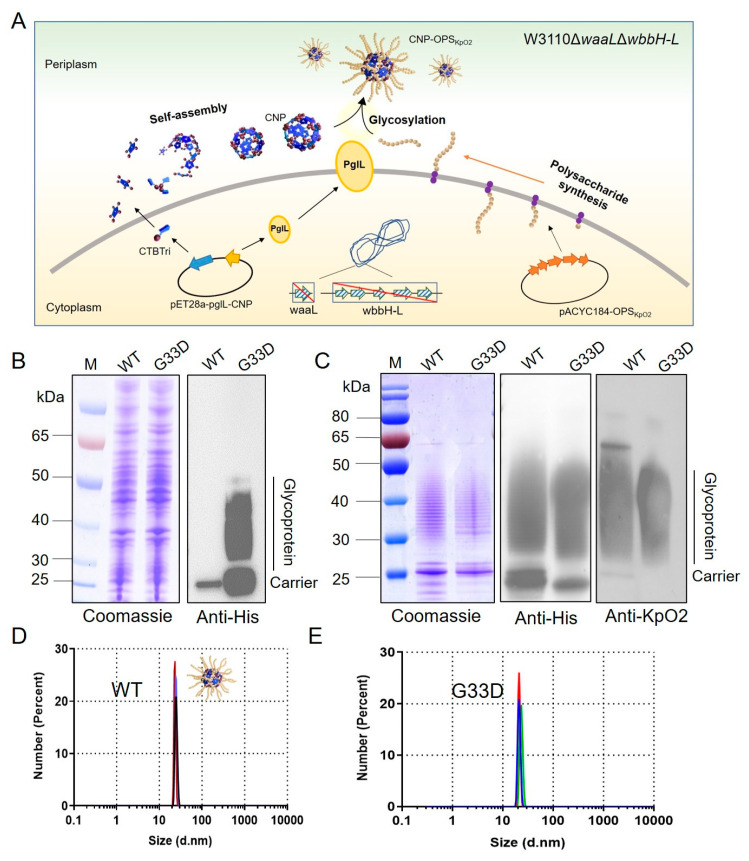
Analysis of the expression of glycoprotein with the G33D mutation. (**A**) Schematic diagram of the biosynthesis of self-assembled conjugated nanovaccines in engineered *E. coli.* (**B**) Detection of glycosylation of the CNP and CNP(G33D) by western blotting with a 6 × His Tag antibody. WT indicates the protein without a mutation. (**C**) Purified CNP(G33D)-OPS_KpO2_ and CNP-OPS_KpO2_ samples were separated by SDS-PAGE and analyzed by Coomassie blue staining and western blotting with a 6 × His tag antibody and KpO2 serum. (**D**) Dynamic light scattering analysis of CNP-OPS_KpO2_. (**E**) Dynamic light scattering analysis of CNP(G33D)-OPS_KpO2_.

**Figure 2 nanomaterials-14-00728-f002:**
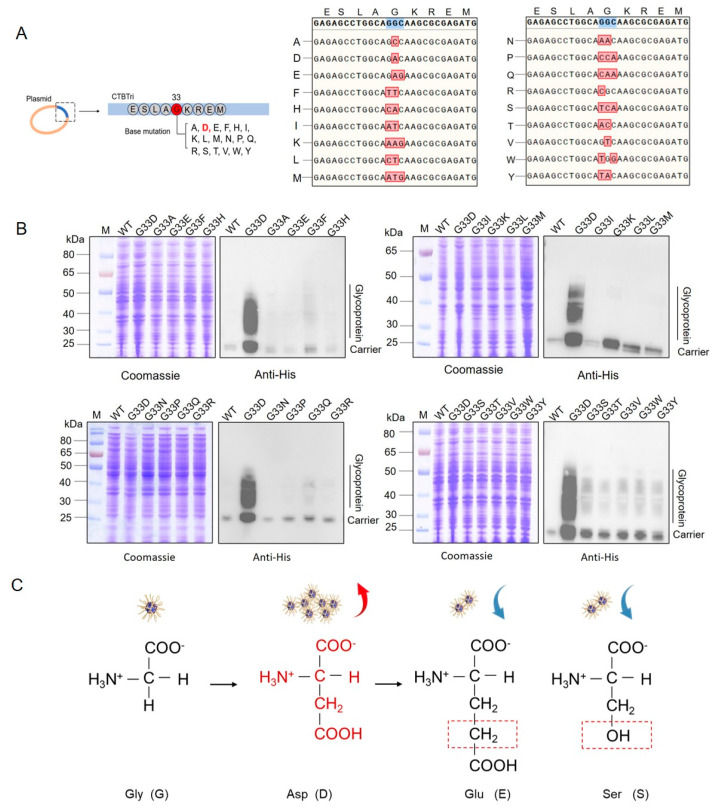
Analysis of the glycosylation of CNP with various mutations at the 33rd position. (**A**) Schematic diagram of the mutations and sequencing results of different base mutations. (**B**) Glycosylation in the W3110ΔΔ/KpO2 strain was detected by western blotting. (**C**) Comparison of amino acids G, D, E, and S.

**Figure 3 nanomaterials-14-00728-f003:**
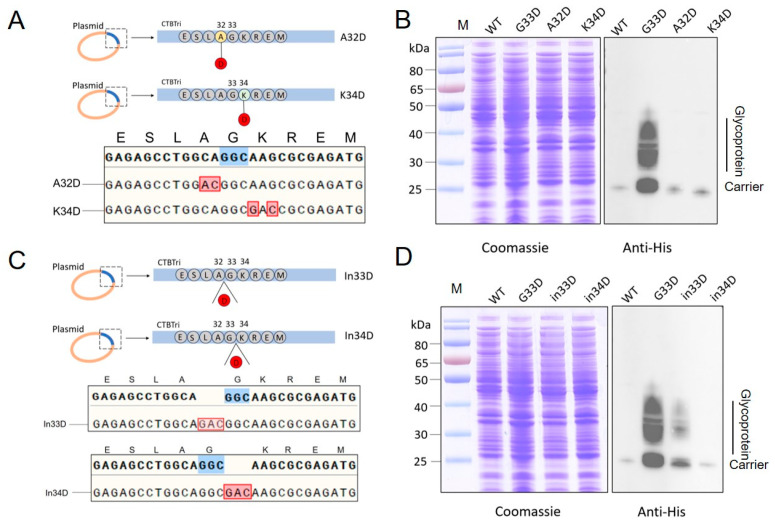
Analysis of glycosylation by mutating animo acids to D around the 33rd position. (**A**) Schematic diagram and sequencing results of A32D and K34D mutations. (**B**) Analysis of the glycosylation of CNP(A32D) and CNP(K34D) by western blotting with a 6 × His tag antibody. (**C**) Schematic diagram and sequencing results of in33D and in34D mutations. (**D**) Glycosylation of CNP(in33D) and CNP(in34D) in the W3110ΔΔ/KpO2 strain was detected by western blotting with a 6 × His Tag antibody.

**Figure 4 nanomaterials-14-00728-f004:**
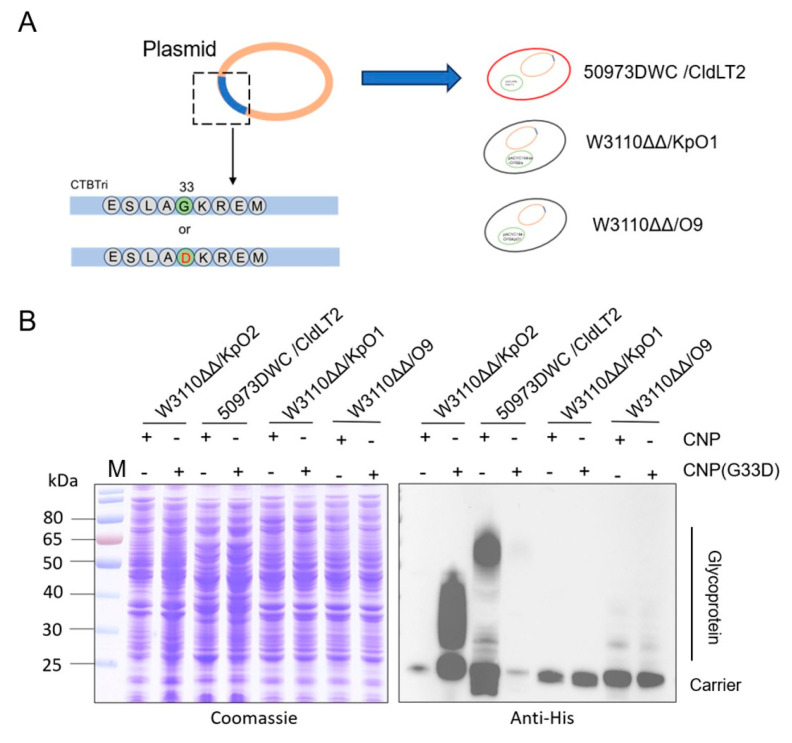
Analysis of glycosylation in various pathogens. (**A**) Schematic diagram of strain construction and mutations. (**B**) Glycosylation in W3110ΔΔ/KpO2, 50973DWC/CldLT2, W3110ΔΔ/O1, and W3110ΔΔ/O9 were detected by western blotting with a 6 × His Tag antibody.

**Figure 5 nanomaterials-14-00728-f005:**
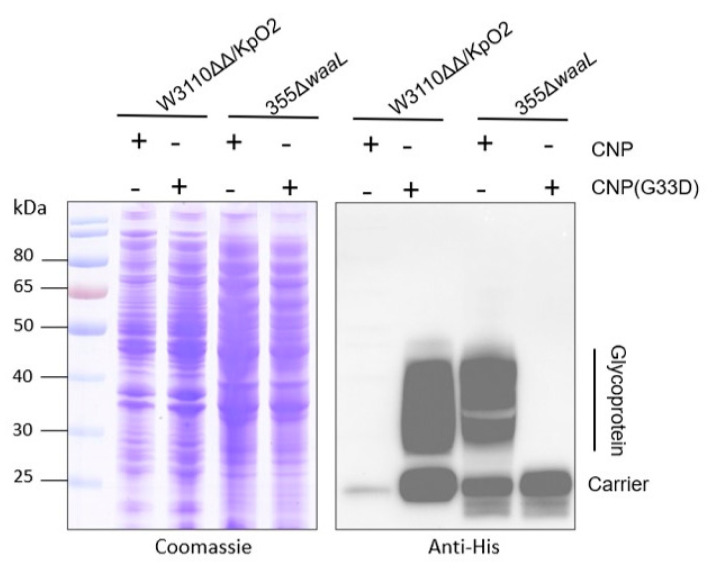
Glycosylation of CNP(G33D) was detected in Kp355. pET28a-pglL-CNP or pET28a-pglL-CNP(G33D) was transformed into Kp355Δ*waaL* and W3110ΔΔ/KpO2, and glycosylation was detected by western blotting with a 6 × His tag antibody.

**Figure 6 nanomaterials-14-00728-f006:**
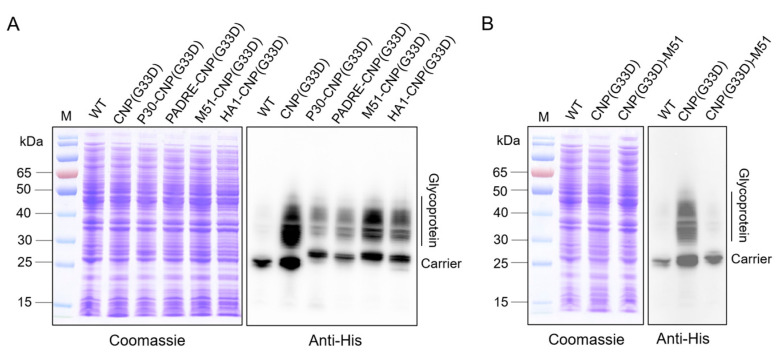
Analysis of the glycosylation of CNP(G33D) by fusing with various T-cell epitopes. (**A**) T-cell epitopes, including P30, PADRE, M51, and HA1 were fused to the N terminus of CNP(G33D), and the glycosylation was analyzed by western blotting with a 6 × His tag antibody. (**B**) M51 was fused at the C terminus of CNP(G33D), and glycosylation was detected by western blotting with a 6 × His tag antibody.

**Figure 7 nanomaterials-14-00728-f007:**
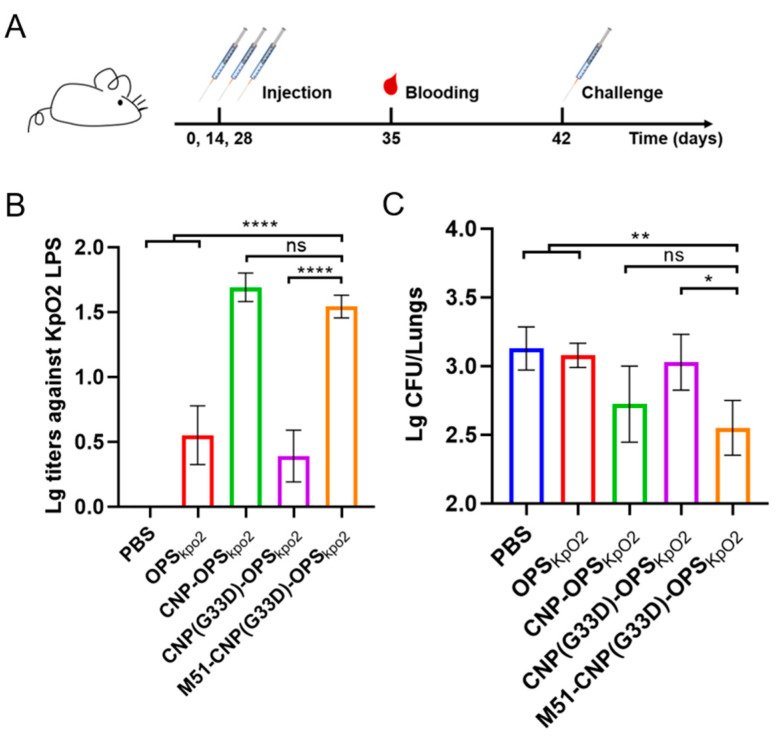
Evaluation of vaccine protection in mouse experiments. (**A**) Process diagram of the vaccine immunization in mice. (**B**) IgG titers against Kp355 LPS in the serum. BALB/c mice were immunized with PBS, CNP-OPS, CNP(G33D)-OPS, or M51-CNP(G33D)-OPS on days 0, 14, and 28, and the serum were sampled 7 days after the third injection (day 35). (**C**) Fourteen days after the final immunization, each mouse was challenged with 2.5 × 10^7^ CFU of Kp355, and 2 days post-injection, the CFUs of Kp355 in the lungs were determined. Data are represented as mean ± s.d. Each group was compared using one-way ANOVA with Dunnett’s multiple-comparison test: **** *p* < 0.0001, ** *p* < 0.01, * *p* < 0.05, and ns > 0.05.

**Figure 8 nanomaterials-14-00728-f008:**
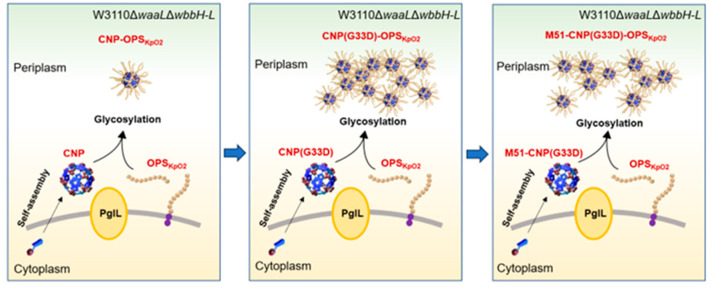
Schematic diagram of the relationship between vector mutations and vaccine yield.

**Table 1 nanomaterials-14-00728-t001:** Plasmids used in this study.

Plasmid	Characteristic	Source
pACYC184-OPS_KpO1_	Encoded O1 serotype OPS of *K. pneumoniae*, Cm^r^	Laboratory stock
pACYC184-OPS_KpO2_	Encoded O2 serotype OPS of *K. pneumoniae*, Cm^r^	Laboratory stock
pET28a-pglL-CNP	Encoded PglL and CNP, Kan^r^	Laboratory stock
pACYC184tac-OPS_Ba_	Encoded OPS of YeO9, Cm^r^	Laboratory stock
pET28a-pglL-CNP(G33D)	Encoded PglL and CNP(G33D), Kan^r^	This work

## Data Availability

The data presented in this study are available on request from the corresponding author.

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
