# Peer review of "Efficient Production of Self-Assembled Bioconjugate Nanovaccines against Klebsiella pneumoniae O2 Serotype in Engineered Escherichia coli"

_nanomaterials, 2024, doi:10.3390/nano14080728_

Round 1
Reviewer 1 Report
Comments and Suggestions for Authors
The manuscript presents an interesting study with potential implications for the field of vaccine development. Addressing the points mentioned below could further strengthen the manuscript and its contribution to the scientific community.
Comments
In the Methods section, please provide references for each cited method. If the method was newly developed in the lab, state the procedure in detail. For example, Table I may not be necessary, and the construction of the new plasmid "pET28a-pglL-CNP(G33D)" should be detailed.
Revise the subtitles of the Methods section to make them more meaningful. For instance, change "Growth Conditions" to "Bacterial Culture," etc.
While the manuscript is well-written, some grammar errors were found.
· Ensure proper use of commas for clarity and to separate items in a list.
· Use consistent tense throughout the manuscript.
· Check for hyphenation errors (e.g., "drug resistance" instead of "drug re-sistance").
· Ensure proper spacing between words and after punctuation marks.
These are just a few examples. A thorough review of the entire manuscript is recommended to identify and correct any additional grammar errors.
Comments on the Quality of English LanguageSee above
Reviewer 2 Report
Comments and Suggestions for Authors
-To what extent does the article discuss the present state of affairs regarding the development of self-assembled bioconjugate nanovaccines that protect against the O2 serotype of Klebsiella pneumoniae?
-To what extent did Escherichia coli undergo any particular engineering to facilitate the efficient manufacture of nanovaccines?
-Is it possible for the authors to shed light on how well and how safely the designed nanovaccines fight Klebsiella pneumoniae infections?
-Is there a comparison between the current methods of vaccine manufacturing against Klebsiella pneumoniae and the suggested approach?
How can this study affect future efforts to create vaccines that are more effective against bacterial diseases?
Reviewer 3 Report
Comments and Suggestions for Authors
The authors highlight the use of nanoparticles (NPs) for the development of vaccines based on the cholera toxin B subunit (CTB) self-assembling into the nanoparticle (CNP). The described results are promising and the bioconjugate nanovaccines with bacterial polysaccharide (OPS) in vivo show a powerful immune response against Klebsiella pneumoniae - serotype vaccine, against Escherichia coli with a very high yield. The authors implemented a stable modification of the 33rd glycine (G) in CNP to aspartate (D), for increased glycoprotein expression.
Innovatively, the study performed a fusion of T cell epitopes, at the end of CNP(G33D) in animals, the fusion of peptide M51 and a high antibody titer were recorded, i.e. a new original nanovaccine CNP-OPSKpO2 is determined.
1. Some sentences are incorrect, please update: Notably, it was discovered that the conjugate vac- 45 cines, synthesized through coupling polysaccharides with carrier proteins, can convert 46 these TI-antigens into T cell-dependent antigens (TD-antigens)... row....45-46; row 60-61; ...row...80-81; ...row...99-10
2. Parts description of materials and methods- 2.4. Immunoblotting and 2.5. Glycoprotein purification - detailed, to be shortened;
3. The results are presented in detail, descriptively. Schematic representation increases the quality of the presented manuscript;
4. Row 258-261 long, detailed, to be shortened; Row 270-273 long, detailed, to be shortened;
5. Discussion part: enhancing immune responses by adding T cell epitopes, in vaccine development should be reinforced with additional information. A suitable final scheme indicating the innovativeness of the study is also appropriate.
6. Conclussion part to be update.
7. The figures are detailed and justified.
8. The use of literature from the last 15 years is impressive. I suggest updating articles where possible.
Comments on the Quality of English Language
minor
Round 2
Reviewer 3 Report
Comments and Suggestions for Authors
-
Author Response
Dear Editor,
Thank you for giving us the opportunity to strengthen our manuscript. We hope both you and the reviewers will be satisfied with the revisions and find that the revised manuscript meets the standards required for publication in Nanomaterials.
Thank you again for your work.
Yours Sincerely,
Chao Pan
State Key Laboratory of Pathogen and Biosecurity,
Beijing Institute of Biotechnology.
Email: panchao@bmi.ac.cn